# Proteome-wide prediction of mode of inheritance and molecular mechanism underlying genetic diseases using structural interactomics

## Abstract

Genetic diseases can be classified according to their modes of inheritance and their underlying molecular mechanisms. Autosomal dominant disorders often result from DNA variants that cause loss-of-function, gain-of-function, or dominant-negative effects, while autosomal recessive diseases are primarily linked to loss-of-function variants. In this study, we introduce a graph-of-graphs approach that leverages protein-protein interaction networks and high-resolution protein structures to predict the mode of inheritance of diseases caused by variants in autosomal genes, and to classify dominant-associated proteins based on their functional effect. Our approach integrates graph neural networks, structural interactomics and topological network features to provide proteome-wide predictions, thus offering a scalable method for understanding genetic disease mechanisms.

## 1 Introduction

Human genetic diseases result from variants that disrupt protein function through diverse molecular mechanisms, which play a critical role in determining their mode of inheritance (MOI) (Zschocke et al., 2023). In autosomal dominant (AD) disorders, a single copy of a mutated gene can result in disease, often through loss of function (LOF) due to haploinsufficiency (HI), where the remaining wild-type allele cannot compensate for the lost function (Veitia, 2002). Dominant disorders can also result from non-LOF mechanisms, such as gain of function (GOF), where the mutant protein acquires a new or altered function, and the dominant-negative (DN) effect, where the mutant isoform interferes with the normal function of the wild-type protein (Backwell & Marsh, 2022). In contrast, autosomal recessive (AR) disorders require variants in both gene copies, predominantly involving LOF mechanisms, such as missense variants that destabilize protein structure or nonsense variants leading to truncated, non-functional proteins.

Previous studies on MOI prediction have introduced computational tools such as DOMINO (Quinodoz et al., 2017), which utilizes linear discriminant analysis (LDA) to predict whether a protein is associated with AD disorders by integrating various features such as genomic data, conservation, and protein interactions. MOI-Pred (Petrazzini et al., 2021), on the other hand, focuses on variant-level predictions, specifically targeting missense variants associated with AR diseases.

More recent research has aimed at predicting the functional impact of variants in specific genes. LoGoFunc combines gene-, protein-, and variant-level features to predict pathogenic GOF, LOF, and neutral variants (Stein et al., 2023). Another study explored the structural effects of variants, finding that non-LOF variants tend to have milder impacts on protein structure (Gerasimavicius et al., 2022). Additionally, a recent study employed three support vector machines (SVM) to predict protein coding genes associated with DN, GOF, and HI mechanisms (Badonyi & Marsh, 2024).

In this study, we present a comprehensive approach for predicting the MOI for all proteins encoded by autosomal genes, as well as elucidating the functional effect of variants underlying AD genetic disorders (Figure 1). Our framework combines graph neural networks (GNNs) (Zhou et al., 2021) with structural interactomics by creating a graph-of-graphs (D'Agostino & Scala, 2014), utilizing both protein-protein interaction (PPI) network and high-resolution protein structures. For MOI pre-

diction, we model proteins as nodes within the PPI network, incorporating topological and protein-level features for classification. For molecular mechanism prediction, we represent each protein as a graph of amino acid residues, leveraging structure-based features to classify the functional effect as HI, GOF, or DN. This integrated approach enables proteome-wide prediction of inheritance patterns and provides mechanistic insights into AD diseases, offering a novel, scalable framework for understanding genetic disorders.

For the sake of flow and conciseness, we refer to "proteins associated with a autosomal dominant (recessive) disorders" as AD (AR) proteins. Similarly, we use DN (GOF/LOF) proteins instead of "proteins associated with DN (GOF/LOF) molecular disease mechanisms".

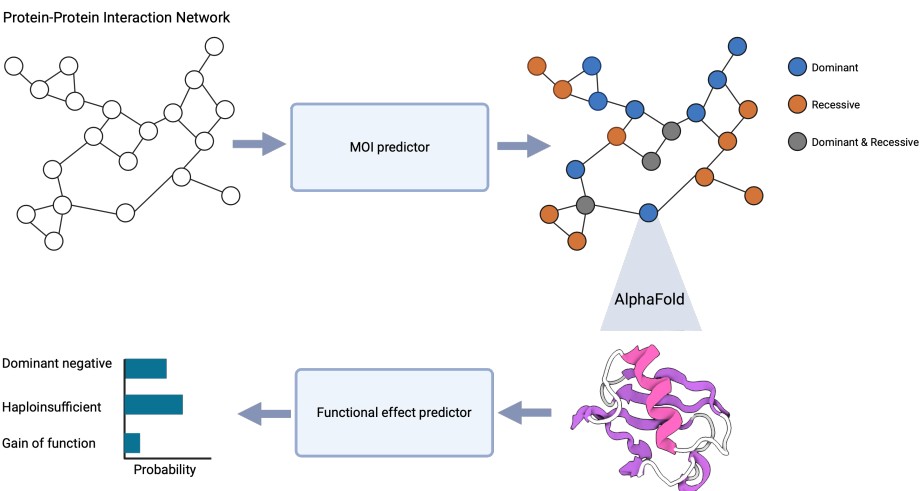

Figure 1: Overview of the study: at first the mode of inheritance (MOI) is predicted for all of the autosomal proteins in the protein-protein interaction network. Afterwards, AlphaFold protein structures are used to generate residue graphs for each dominant protein, and functional effects are predicted based on these graphs. Figure created with `BioRender.com`.

## 2 METHODS

### 2.1 DATA COLLECTION

**Mode of inheritance**  We collected the MOI data from the Gene Curation Coalition (GenCC) (DiStefano et al., 2022) as well as the Online Mendelian Inheritance in Man (OMIM) (Hamosh, 2002). For GenCC records, we kept records with definitive, strong, or moderate gene-disease clinical validity. We focused on autosomal proteins, due to intrinsic differences in MOI for X chromosome proteins. Proteins were accordingly labeled as AD, AR, or ADAR (both dominant and recessive).

**Molecular mechanism**  We collected the functional effect of AD proteins from Badonyi & Marsh (2024). This is a curated set of AD proteins labeled with their known functional effects, including DN, GOF, and HI.

**PPI network**  To make a comprehensive PPI network, we combined the interaction from four resources: STRINGdb with interaction score $\geq 0.7$ (Szklarczyk et al., 2022), BioGRID (Oughtred et al., 2020), the Human Reference Interactome (HuRI) (Luck et al., 2020), and Menche et al. (2015), which resulted in a network with 17,248 nodes, and 375,494 edges.

**Protein graph**  We downloaded the predicted structures of all human proteins from the AlphaFold database (Varadi et al., 2023). We then used Graphein (Jamasb et al., 2022) to construct a residue graph per protein based on the protein structures. In such residue graphs, nodes are amino acids and edges are various interaction between them, including peptide bonds, aromatic interaction, hydrogen bonds, disulfide bonds, ionic interactions, aromatic-sulfur interactions, and cation-$\pi$ interactions.

**Protein features**   We annotated all proteins with 78 features. Based on their definition, we clustered the features into three groups: 1) structure and function 2) conservation and constraint 3) expression and regulation. The complete list of the protein features is available at A.1.

**Residue features**   For the residue graphs, we annotated the nodes (i.e. amino acids) with 73 features. We grouped them into four clusters based on their description: 1) structure and function 2) sequence 3) biochemical 4) evolutionary. The complete description of residue features can be found in A.2.

## 2.2 MODEL DEVELOPMENT

**Study design**   In this study, MOI is predicted by classifying PPI network nodes, while functional effect prediction is performed as a graph classification task. In both models, we considered multi-label classification, where inputs can have more than one label. For all the following steps, we used PyTorch Geometric library (Fey & Lenssen, 2019).

**Architecture**   For both MOI and functional effect prediction, we utilized various graph neural network architecture including graph convolutional network (GCN) (Kipf & Welling, 2017), graph attention network (GAT) (Brody et al., 2022), and graph isomorphism network (GIN) (Xu et al., 2019).

GCNs extend the concept of convolution from grid-like data (such as images) to graph data, allowing the aggregation of feature information from neighboring nodes. This approach effectively captures local graph structure and node features. The forward propagation formula in a GCN is given by:

$$h_i^{(l+1)} = \sum_{j \in \mathcal{N}(i)} \frac{1}{\sqrt{\deg(i)}\sqrt{\deg(j)}} \mathbf{W}^{(l)} h_j^{(l)}$$

- $h_j^{(l)}$: The node feature vector at layer $l$.

- $h_i^{(l+1)}$: The updated node feature vector at layer $l + 1$.

- $\mathbf{W}^{(l)}$: The learnable weight matrix for layer $l$.

- $\mathcal{N}(i)$: The set of neighbors of node $i$ (including itself due to the self-loop).

- $\frac{1}{\sqrt{\deg(i)}\sqrt{\deg(j)}}$: The normalization term based on the degrees of nodes $i$ and $j$, ensuring that nodes with different degrees contribute proportionally to the update.

GINs are designed to be powerful for graph isomorphism, making them capable of distinguishing a wide variety of graph structures. They achieve this by using a multi-layer perceptron (MLP) to aggregate node features, enhancing their discriminative power. The update rule for the GIN is given by:

$$h_i^{(l+1)} = \text{MLP}^{(l)} \left( \left(1 + \epsilon^{(l)}\right) h_i^{(l)} + \sum_{j \in \mathcal{N}(i)} h_j^{(l)} \right)$$

- $h_i^{(l)}$: The node feature vector at layer $l$.

- $h_i^{(l+1)}$: The updated node feature vector at layer $l + 1$.

- $\text{MLP}^{(l)}$: A multi-layer perceptron applied at layer $l$, which acts as a learnable transformation function on the aggregated node features.

- $\epsilon^{(l)}$: A learnable parameter at layer $l$ that adjusts the contribution of the central node's own features $h_i^{(l)}$.

- $\mathcal{N}(i)$: The set of neighbors of node $i$. The sum $\sum_{j \in \mathcal{N}(i)} h_j^{(l)}$ aggregates the features of all neighbor nodes in layer $l$.

GATs introduce attention mechanisms to GNNs, enabling nodes to assign different importance weights to their neighbors. This allows for more flexible and expressive feature aggregation, potentially improving performance on tasks where certain neighbors have more influence than others. The forward propagation rule for GAT is given by:

$$h_i^{(l+1)} = \sigma \left( \sum_{j \in \mathcal{N}(i)} \alpha_{ij}^{(l)} \mathbf{W}^{(l)} h_j^{(l)} \right)$$

$$\alpha_{ij}^{(l)} = \frac{\exp \left( \text{LeakyReLU} \left( a^T \left[ \mathbf{W}^{(l)} (h_i^{(l)} \| h_j^{(l)}) \right] \right) \right)}{\sum_{k \in \mathcal{N}(i)} \exp \left( \text{LeakyReLU} \left( a^T \left[ \mathbf{W}^{(l)} (h_i^{(l)} \| h_k^{(l)}) \right] \right) \right)}$$

- $h_i^{(l)}$: The node feature vector at layer $l$.

- $h_i^{(l+1)}$: The updated node feature vector at layer $l + 1$.

- $\alpha_{ij}^{(l)}$: The attention coefficient between nodes $i$ and $j$.

- $\mathbf{W}^{(l)}$: The weight matrix at layer $l$.

- $a$: The learnable attention vector.

- $\|$: The concatenation operator.

- $\mathcal{N}(i)$: The set of neighbors of node $i$.

- $\sigma(\cdot)$: A non-linear activation function (ReLU in our implementation).

In all the models, we used 2 hidden layers with 128 and 64 units. The output layer dimension is two for MOI models (AD and AR), and three for functional effect models (DN, HI, and GOF). We used dropout (Srivastava et al., 2014) and weight decay (Loshchilov & Hutter, 2019) to mitigate the chance of over-fitting.

**Training and evaluation** We trained each model using a binary cross entropy loss on $80\%$ of the data for maximum 100 epochs, and used early stopping based on validation loss to avoid overfitting. We evaluated each selected model on $10\%$ of the unseen test data using $F_1$, precision, and recall scores.

We benchmarked the performance of our model against previous state-of-the-art approaches. For MOI prediction, we compared our model with DOMINO (Quinodoz et al., 2017), which predicts the probability of a protein's association with dominant disorders (pAD). We used our MOI test set and excluded any proteins present in DOMINO's training data. Since no threshold was provided, we classified proteins as AD if pAD $> 0.6$, AR if pAD $< 0.4$, and ADAR otherwise.

For functional effect prediction, we compared our model with the models from Badonyi & Marsh (2024), which include three separate SVM models (DN vs LOF, GOF vs LOF, and LOF vs non-LOF). We combined the test sets from these models and used the pre-calculated probabilities to evaluate performance in a multi-label classification setting.

**Explanation** To study the importance of features, we utilized Integrated Gradients (Sundararajan et al., 2017) using Captum (Kokhlikyan et al., 2020). Since this method works per sample, we applied it on correctly predicted samples in the test sets. We included samples with only one label for further interpretability. Finally, we averaged feature attributions across selected samples, and scaled them by dividing to the maximum attribution.

## 2.3 PROTEOME-WIDE INFERENCE

**MOI and molecular mechanism inference** After selecting the final models for MOI and functional effect prediction, we predicted the MOI for all proteins in the PPI network. Afterwards, we predicted the functional effect for the subset of proteins that were predicted as AD or ADAR.

**Enrichment analysis** To study further the predictions, we used GSEApy (Fang et al., 2022) to perform enrichment analysis (Khatri et al., 2012), which is a statistical method used to determine whether known biological functions or processes are over-represented in a protein list of interest (e.g. AD proteins). In this method, the enrichment significance is calculated based on the hypergeometric distribution, where p-value is the cumulative probability of observing at least $k$ proteins of interest annotated to a specific protein set. The formula for the p-value is given by:

$$p = 1 - \sum_{i=0}^{k-1} \frac{\binom{M}{i}\binom{N-M}{n-i}}{\binom{N}{n}},$$

where $N$ is the total number of proteins in the background distribution, $M$ is the number of proteins in that distribution annotated to the gene set of interest, $n$ is the size of the list of proteins of interest, and $k$ is the number of proteins in that list which are annotated to the gene set.

For proteins predicted as only AD or AR, we used DisGeNET (Piñero et al., 2019) as reference to investigate the enrichment of AD or AR proteins in certain diseases. For AD proteins predicted as DN, HI, or GOF, we used Gene Ontology (Ashburner et al., 2000; Aleksander et al., 2023) to understand their functional landscape.

## 3 RESULTS

### 3.1 DATASETS

**MOI data** We gathered 4,737 MOI-labeled proteins, among them 2,494 (53%) were only AR, 1,420 (30%) were only AD, and 808 (17%) were both AD and AR (Figure 2, left).

**Functional effect data** We collected 1,276 proteins with annotated functional effect, among them 250 (20%) were only DN, 376 (29%) were only HI, 251 (20%) were only GOF, 114 (9%) were both DN and HI, 115 (9%) were both DN and GOF, 92 (7%) were both HI and GOF, and 78 (6%) were all of the DN, HI, GOF (Figure 2, right).

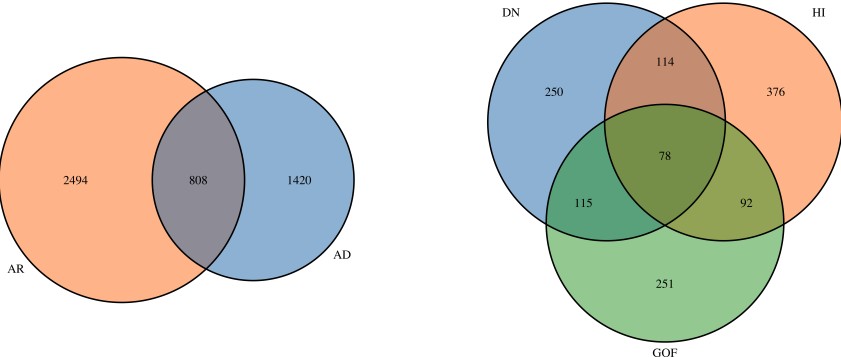

Figure 2: The number of proteins with labeled MOI (left) and molecular mechanism (right).

### 3.2 MODELS PERFORMANCE EVALUATION

**MOI models** We evaluated all trained models on the unseen test set (Table 1). The GCN model achieved the highest precision score, while the GAT model had the best recall, with both models yielding an $F_1$ score of 0.74. Due to the class imbalance in the MOI dataset, we prioritized maximizing recall and therefore selected the GAT model. We also assessed the performance of DOMINO (Quinodoz et al., 2017) as outlined in the methods section (2.2), and found that our models outperformed it (Table 1).

Table 1: MOI prediction performance on the test set

| Metric | GCN | GAT | GIN | LDA (Quinodoz et al., 2017) |
|---|---|---|---|---|
| F1 | **0.74** | **0.74** | 0.71 | 0.71 |
| Precision | **0.77** | 0.75 | 0.76 | 0.76 |
| Recall | 0.73 | **0.74** | 0.66 | 0.67 |

Table 2: Functional effect prediction performance on the test set

| Metric | GCN | GAT | GIN | SVM (Badonyi & Marsh, 2024) |
|---|---|---|---|---|
| F1 | **0.61** | 0.49 | 0.57 | 0.59 |
| Precision | 0.58 | 0.59 | 0.57 | **0.67** |
| Recall | **0.67** | 0.43 | 0.63 | 0.54 |

**Functional effect models**  Table 2 shows the performance of various models on the functional effect test set, with the GCN model achieving the highest $F_1$ and recall scores. We also evaluated the SVM models from Badonyi & Marsh (2024) as described in the methods section (2.2). Based on the overall performance, we selected the GCN model as the final model for functional effect prediction.

### 3.3 MODELS INTERPRETATION

**MOI feature attribution**  Using the GAT model, we calculated features attribution separately for correctly predicted AD or AR proteins in the test set.

We observed that the most important predictors for AD prediction are features related to constraint and conservation (Figure S1). The top feature was pLI, which is probability of loss-of-function intolerance (Lek et al., 2016). Using the labeled data, we observed that AD proteins have higher pLI values compared to AR proteins (Figure 3).

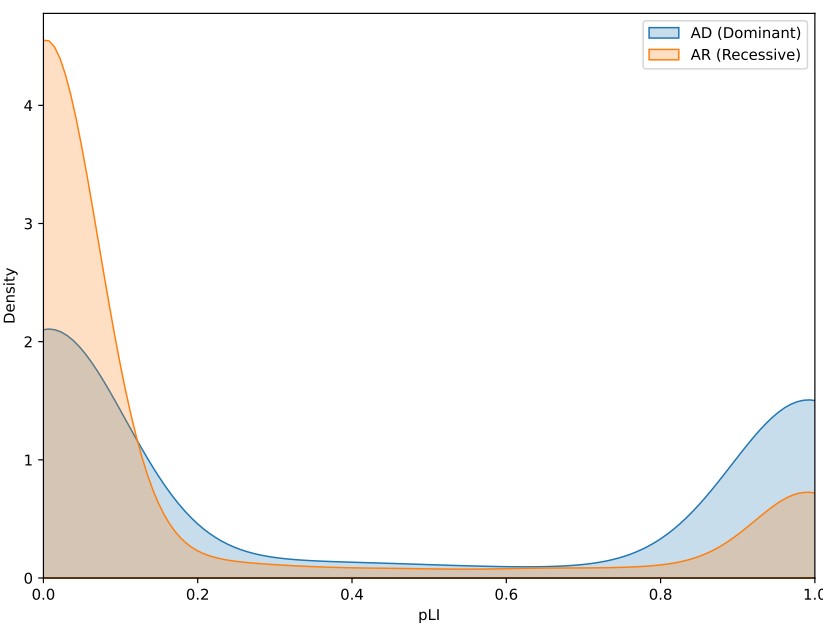

Figure 3: pLI distribution for AD and AR genes.

For AR prediction, the most important feature was localization inside mitochondria (Figure S2). Using the ground truth dataset, we observed that AR proteins are more likely to be localized inside mitochondria compared to AD proteins ($OR = 3.13, CI = [2.47, 3.97]$) (Figure 4).

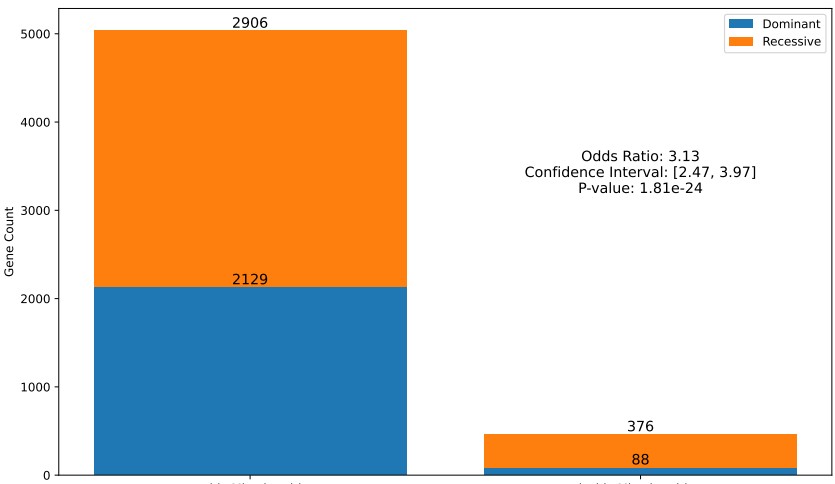

Figure 4: Number of proteins with sub-cellular localization inside or outside mitochondria. The odds ratio was calculated as $\left(\frac{\text{AR\_inside}}{\text{AR\_outside}}\right) \big/ \left(\frac{\text{AD\_inside}}{\text{AD\_outside}}\right)$. P-value was calculated using the Fisher's exact test.

**Functional effect feature attribution**    Using the GCN model, we measured features attribution for correctly predicted DN, HI, and GOF proteins. Because features are at residue-level and prediction are at protein-level, we cannot draw direct conclusions from these measurements, yet they can help to understand the associations.

For DN proteins, the most important feature was the MoRFchibi score (Malhis et al., 2016) (Figure S3), which predicts Molecular Recognition Features (MoRFs). MoRFs are disordered regions that fold upon binding with other peptides and proteins.

For HI proteins, as shown in Figure S4, the presence of topological domains is the strongest predictor. This feature was derived from UniProt (Bateman et al., 2022).

Feature attribution analysis for GOF proteins showed that top feature is the molar fraction of 20 amino acids in samples of 2001 buried residues, derived from Janin (1979) using the ExPASy ProtScale (Gasteiger, 2003).

### 3.4    PROTEOME-WIDE INFERENCE

**MOI prediction for all autosomal proteins**    Out of 17,248 nodes on the PPI network, 16,184 (94%) were autosomal, and we used the GAT model to predict the most likely MOI for all of them. 7,871 (49%) of them were predicted to be AR, 6,862 (42%) were predicted to be AD, and 1451 (9%) were predicted to be ADAR (Figure S6). As expected, we observed a strong negative correlation between the probability of being AD and AR (Pearson correlation coefficient = -0.96) (Figure 5). Finally, we performed pathway enrichment analyses for AD and AR proteins separately. AD proteins were significantly enriched in various cancers (Figure S7), while AR proteins were significantly over-represented in mitochondrial and neuro-developmental disorders (Figure S8).

**Functional effect prediction for all AD-predicted proteins**    Based on the proteome-wide MOI predictions, we identified 8,313 AD or ADAR proteins, and predicted their functional effect using the GCN model. Among them, 450 (5%) were only DN, 2,155 (26%) were only HI, 415 (5%) were only GOF, 3,610 (43%) were both DN and HI, 757 (9%) were both DN and GOF, 802 (10%) were both HI and GOF, and 72 (1%) were DN, HI and GOF (Figure S9). Pathway enrichment analysis revealed that DN proteins are enriched in pathways associated with filament organization (Figure S10), HI proteins are over-represented in pathways related to transcription regulation and cell cycle

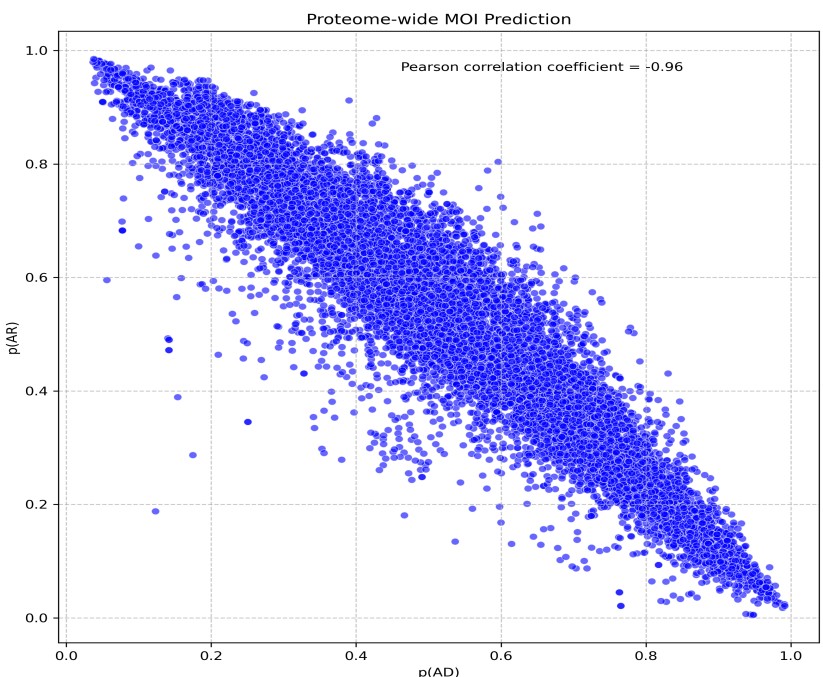

Figure 5: probability of AD (pAD) vs probability of AR (pAR) for all autosomal proteins.

control (Figure S11), and GOF proteins were enriched in pathways related to ion transport (Figure S12).

# 4    DISCUSSION

In this work, we introduce a novel framework that integrates GNNs with structural interactomics to predict both the MOI and the functional effect of mutated proteins in genetic disorders. By leveraging PPI network and high-resolution protein structures, we offer a graph-of-graphs approach that addresses two critical aspects of genetic disease prediction. This allows us to not only classify proteins as AD or AR but also predict whether AD diseases manifest through HI, GOF, or DN mechanisms.

Our framework demonstrated good performance in predicting MOI, with the GAT model achieving the best recall for identifying AD and AR proteins. Notably, we found that proteins predicted as AD were strongly enriched in cancer pathways, while AR proteins were predominantly associated with mitochondrial and neurodevelopmental disorders. In terms of functional effects, the GCN model effectively classified HI, GOF, and DN proteins based on structural features. Feature attribution analysis revealed that DN proteins were associated with high MoRFchibi scores (Malhis et al., 2016), which might indicate regions involved in protein-protein interactions, potentially at interfaces. HI proteins were linked to the presence of topological domains, while GOF proteins were associated with features related to the amino acid composition of buried residues.

While our approach offers a comprehensive view of inheritance patterns and functional effects, there are several limitations. First, the availability of high-quality structural data for all human proteins is still limited, which could restrict the accuracy of our predictions (Bertoline et al., 2023). Additionally, our reliance on existing PPI network data may introduce biases, as not all interactions are equally well-characterized across different tissues or biological contexts (Ziv et al., 2022). Furthermore, the imbalance in labeled training data may impact the model performance on these classes. Finally, although our method captures the functional effect of AD proteins, it does not extend to other modes of inheritance or interactions that may occur at a multi-variant or epistatic level (Phillips, 2008).

Moving forward, there are several avenues for expanding this work. First, incorporating tissue-specific PPI networks and expression data could enhance the precision of our predictions, especially for proteins with context-dependent functions (Ziv et al., 2022). Additionally, expanding the model to account for more complex inheritance patterns, such as polygenic traits and epistasis, could provide a more comprehensive understanding of genetic disease (Boyle et al., 2017). Finally, improving the interpretability of models in biological contexts remains essential to derive more actionable insights from the predictions (Chen et al., 2024b).

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

# A APPENDIX

## A.1 PROTEIN FEATURES DESCRIPTION

| Protein Structure and Function | Description |
| --- | --- |
| PSIPRED_helix (Basu et al., 2024) | Prediction of helical secondary structures. |
| PSIPRED_strand (Basu et al., 2024) | Prediction of beta-strand secondary structures. |
| ASAquick_buried (Basu et al., 2024) | Prediction of buried surface area (solvent accessibility). |
| flDPnn_disorder (Basu et al., 2024) | Prediction of intrinsically disordered regions. |
| MoRFchibi_morf (Basu et al., 2024) | Prediction of molecular recognition features (MoRFs). |
| DFLpred_linker (Basu et al., 2024) | Prediction of disordered flexible linker residues. |

| DisoRDPbind_RNA (Basu et al., 2024) | Prediction of RNA-binding disordered regions. |
|---|---|
| DisoRDPbind_DNA (Basu et al., 2024) | Prediction of DNA-binding disordered regions. |
| DisoRDPbind_PRO (Basu et al., 2024) | Prediction of protein-binding disordered regions. |
| DRNApred_RNA (Basu et al., 2024) | Prediction of RNA-binding residues. |
| DRNApred_DNA (Basu et al., 2024) | Prediction of DNA-binding residues. |
| SignalP (Basu et al., 2024) | Prediction of signal peptides. |
| SCRIBER_PRO (Basu et al., 2024) | Prediction of protein-binding residues. |
| PTM_content (Basu et al., 2024) | Prediction of post-translational modification sites. |
| membrane_propensity (Badonyi & Marsh, 2024) | Propensity for membrane association. |
| Plastid (Bateman et al., 2022) | Localization to plastid. |
| CellMembrane (Bateman et al., 2022) | Localization to cell membrane. |
| Cytoplasm (Bateman et al., 2022; Saadat & Fellay, 2024b) | Localization to cytoplasm. |
| EndoplasmicReticulum (Bateman et al., 2022) | Localization to endoplasmic reticulum. |
| Extracellular (Bateman et al., 2022) | Localization to extracellular space. |
| GolgiApparatus (Bateman et al., 2022) | Localization to Golgi apparatus. |
| LysosomeOrVacuole (Bateman et al., 2022) | Localization to lysosome or vacuole. |
| Mitochondrion (Bateman et al., 2022) | Localization to mitochondrion. |
| Nucleus (Bateman et al., 2022) | Localization to nucleus. |
| Peroxisome (Bateman et al., 2022) | Localization to peroxisome. |
| MembraneBound (Bateman et al., 2022) | Membrane-bound proteins. |
| aco (Badonyi & Marsh, 2024) | Absolute contact order of the protein structure. |
| pct_buried (Badonyi & Marsh, 2024) | Fraction of buried residues in protein structure. |
| plddt (Badonyi & Marsh, 2024) | Mean pLDDT confidence score of predicted structures. |
| pi (Badonyi & Marsh, 2024) | Protein isoelectric point. |
| ct (Badonyi & Marsh, 2024) | Cotranslational assembly annotations. |
| efx_abs (Badonyi & Marsh, 2024) | Median ratio of ESM-1v and absolute FoldX $\Delta\Delta G$ for missense mutations. |
| efx_raw (Badonyi & Marsh, 2024) | Median ratio of ESM-1v and raw FoldX $\Delta\Delta G$ for missense mutations. |
| median_scriber (Badonyi & Marsh, 2024) | Median SCRIBER score for residues with more than 5% relative solvent accessible surface area. |

| Evolutionary Conservation and Variation | Description |
|---|---|
| MMseq2_low_conservation (Basu et al., 2024) | Low conservation from MMseqs. |
| MMseq2_high_conservation (Basu et al., 2024) | High conservation from MMseqs. |
| phastCons7way_mean (Zeng et al., 2024) | Mean conservation score across 7 species. |
| phastCons7way_max (Zeng et al., 2024) | 95th percentile conservation score across 7 species. |

| | |
|---|---|
| phastCons17way_max (Zeng et al., 2024) | 95th percentile conservation score across 17 species. |
| phastCons100way_max (Zeng et al., 2024) | 95th percentile conservation score across 100 species. |
| fracCdsPhylopAm (Zeng et al., 2024) | Fraction of coding sequences constrained in 240 mammals. |
| dn_ds (Badonyi & Marsh, 2024) | Human-macaque dN/dS ratio of nonsynonymous to synonymous substitutions. |
| UNEECON_G (Zeng et al., 2024) | Evolutionary pressure score (UNEECON). |
| n_paralogs (Badonyi & Marsh, 2024) | Number of paralogous proteins. |
| max_id (Badonyi & Marsh, 2024) | Maximum sequence identity to paralogs. |
| nc_gerp (Badonyi & Marsh, 2024) | GERP++ score for non-coding regions. |
| phylop_5utr (Zeng et al., 2024) | Evolutionary conservation of 5' UTR. |
| ExAC_don_to_syn (Lek et al., 2016) | Donor to synonymous mutation ratio from ExAC. |
| lof.pLI (Chen et al., 2024a) | Probability of being loss-of-function intolerant. |
| lof.pNull (Chen et al., 2024a) | Null hypothesis for loss-of-function. |
| lof.pRec (Chen et al., 2024a) | Probability of intolerance to homozygous but not heterozygous loss-of-function variants. |
| lof.oe_ci.upper (Chen et al., 2024a) | Upper confidence interval for loss-of-function over-expected score. |
| shet (Zeng et al., 2024) | Selection coefficient related to heterozygosity. |
| mis.z_score (Chen et al., 2024a) | Z-score for missense variation constraint. |
| syn.z_score (Chen et al., 2024a) | Z-score for synonymous variation constraint. |

| Transcripts Expression Regulation | Description |
|---|---|
| abundance (Badonyi & Marsh, 2024) | Protein abundance (from PaxDB). |
| exp_var (Badonyi & Marsh, 2024) | RNA expression variance across tissues. |
| tau (Zeng et al., 2024) | Tissue specificity of gene expression (0, broadly expressed to 1, tissue specific). |
| TF (Zeng et al., 2024) | Indicates if the gene is a transcription factor. |
| EDS (Zeng et al., 2024) | Enhancer domain score. |
| ABC_count1 (Zeng et al., 2024) | Number of biosamples with an active ABC enhancer. |
| ABC_count2 (Zeng et al., 2024) | Total number of ABC enhancers across all biosamples. |
| ABC_count3 (Zeng et al., 2024) | Total number of ABC enhancers after union of enhancer domains. |
| ABC_length_per_type (Zeng et al., 2024) | Average ABC enhancer length per active cell type. |
| Roadmap_count1 (Zeng et al., 2024) | Number of biosamples with an active Roadmap enhancer. |
| Roadmap_count2 (Zeng et al., 2024) | Total number of Roadmap enhancers across all biosamples. |
| Roadmap_count3 (Zeng et al., 2024) | Total number of Roadmap enhancers after union of enhancer domains. |
| promoter_count (Zeng et al., 2024) | Number of promoters. |
| mRNA_halflife_10 (Sharova et al., 2009) | mRNA half-life in hours. |
| CDS_GC (Zeng et al., 2024) | GC content of the coding sequence. |
| UTR3_length (Zeng et al., 2024) | Length of 3' UTR. |
| UTR3_GC (Zeng et al., 2024) | GC content of 3' UTR. |
| UTR5_length (Zeng et al., 2024) | Length of 5' UTR. |
| UTR5_GC (Zeng et al., 2024) | GC content of 5' UTR. |
| transcript_length (Zeng et al., 2024) | Total transcript length. |
| Transcript_count (Zeng et al., 2024) | Number of transcripts. |
| num_exons (Zeng et al., 2024) | Number of exons. |

| connect_decile (Zeng et al., 2024) | Decile rank of connectedness in coexpression networks. |
|---|---|
| connect_quantile (Zeng et al., 2024) | Quantile rank of connectedness in coexpression networks. |
| connectedness (Zeng et al., 2024) | Overall connectedness in coexpression networks. |

## A.2 Residue features description

| Structure and Function | Description |
|---|---|
| STRAND (Bateman et al., 2022; Saadat & Fellay, 2024b) | Beta strand regions in the protein structure. |
| HELIX (Bateman et al., 2022; Saadat & Fellay, 2024b) | Alpha helix regions in the protein structure. |
| COILED (Bateman et al., 2022; Saadat & Fellay, 2024b) | Coiled-coil regions of the protein. |
| PSIPRED_helix (Basu et al., 2024) | Prediction of helical secondary structures. |
| PSIPRED_strand (Basu et al., 2024) | Prediction of beta-strand secondary structures. |
| alpha_helixfasman (Gasteiger, 2003) | Helix propensity based on the Fasman algorithm. |
| beta_turnfasman (Gasteiger, 2003) | Beta turn propensity based on the Fasman algorithm. |
| TOPO_DOM (Bateman et al., 2022; Saadat & Fellay, 2024b) | Topological domains of the protein. |
| TRANSMEM (Bateman et al., 2022; Saadat & Fellay, 2024b) | Transmembrane regions in the protein structure. |
| DOMAIN (Bateman et al., 2022; Saadat & Fellay, 2024a) | Functional/structural domains of the protein. |
| REGION (Bateman et al., 2022; Saadat & Fellay, 2024b) | General regions in the protein. |
| REPEAT (Bateman et al., 2022; Saadat & Fellay, 2024b) | Repetitive sequences in the protein. |
| ZN_FING (Bateman et al., 2022; Saadat & Fellay, 2024b) | Zinc finger domains involved in binding. |
| COMPBIAS (Bateman et al., 2022; Saadat & Fellay, 2024b) | Regions with compositional bias. |
| ACT_SITE (Bateman et al., 2022; Saadat & Fellay, 2024b) | Active sites in the protein. |
| BINDING (Bateman et al., 2022; Saadat & Fellay, 2024b) | Binding sites for ligands, substrates, or other molecules. |
| DISULFID (Bateman et al., 2022; Saadat et al., 2023) | Disulfide bonds stabilizing the protein structure. |
| PROPEP (Bateman et al., 2022; Saadat & Fellay, 2024b) | Propeptide regions that are cleaved during maturation. |
| SIGNAL (Bateman et al., 2022; Saadat & Fellay, 2024b) | Signal peptides for protein targeting. |
| TRANSIT (Bateman et al., 2022; Saadat & Fellay, 2024b) | Transit peptides for directing proteins to organelles. |
| DNA_BIND (Bateman et al., 2022; Saadat & Fellay, 2024b) | DNA-binding regions. |
| DisoDNAscore (Basu et al., 2024) | Propensity for disordered regions to bind DNA. |
| DisoRNAscore (Basu et al., 2024) | Propensity for disordered regions to bind RNA. |
| DisoPROscore (Basu et al., 2024) | Propensity for disordered regions to bind proteins. |
| DRNApredDNAscore (Basu et al., 2024) | Prediction of DNA-binding residues. |
| DRNApredRNAscore (Basu et al., 2024) | Prediction of RNA-binding residues. |
| MoRFchibiScore (Basu et al., 2024) | Prediction of molecular recognition features (MoRFs). |

| SCRIBERscore (Basu et al., 2024) | Prediction of protein-binding residues. |
|---|---|
| hbond_acc | Hydrogen bond acceptor residues. |
| hbond_donor | Hydrogen bond donor residues. |
| c_beta_vector0, c_beta_vector1, c_beta_vector2 | Geometric arrangement of side chains (C-beta vectors). |
| sequence_neighbour_vector_n_to_c0, sequence_neighbour_vector_n_to_c1, sequence_neighbour_vector_n_to_c2 | Sequence neighbors from N- to C-terminus. |

| Sequence | Description |
|---|---|
| aa0 to aa19 | Representation of the 20 standard amino acids. |
| a_a_composition | Amino acid composition. |
| numbercodons (Gasteiger, 2003) | Number of codons coding for each amino acid. |
| ratioside (Gasteiger, 2003) | Ratio of side chain types (e.g., polar vs. nonpolar). |

| Biochemical | Description |
|---|---|
| bulkiness (Gasteiger, 2003) | Bulkiness of amino acid side chains. |
| isoelectric_points (Gasteiger, 2003) | Isoelectric points of residues. |
| averageburied (Gasteiger, 2003) | Average number of buried residues in the protein. |
| buriedresidues (Gasteiger, 2003) | Residues buried within the protein structure. |
| accessibleresidues (Gasteiger, 2003) | Solvent-accessible residues in the protein. |
| ASAquick_normscore (Basu et al., 2024) | Normalized accessible surface area score. |
| hphob_argos (Gasteiger, 2003) | Hydrophobicity score from the Argos scale. |
| hphob_welling (Gasteiger, 2003) | Hydrophobicity score from the Welling scale. |
| flDPnn_score (Basu et al., 2024) | Prediction of disorder regions from flDPnn. |
| DFLpredScore (Basu et al., 2024) | Prediction of disordered flexible linkers. |
| averageflexibility (Gasteiger, 2003) | Average flexibility of residues. |

| Evolutionary | Description |
|---|---|
| MMseq2_conservation_score (Basu et al., 2024) | Conservation score based on MMseq2. |
| relativemutability (Gasteiger, 2003) | Likelihood of amino acid mutation over evolutionary time. |

## A.3 SUPPLEMENTARY FIGURES

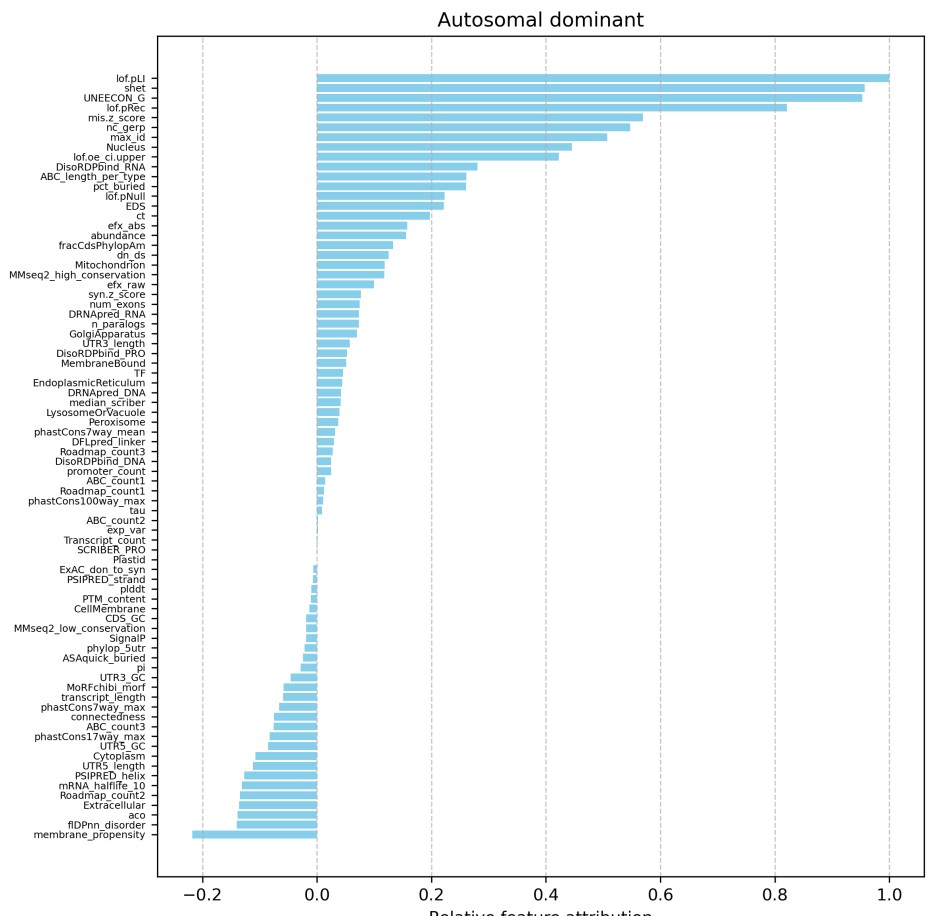

Supplementary Figure S1: GAT model interpretation for AD prediction.

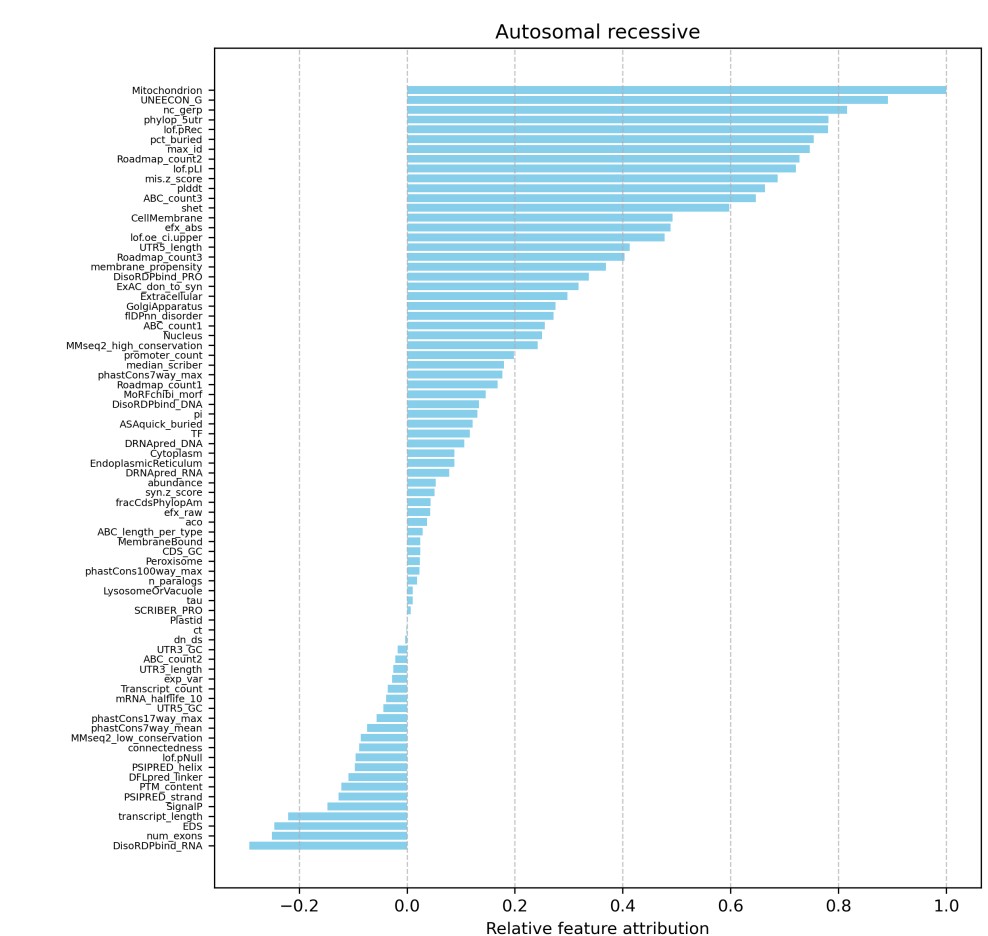

Supplementary Figure S2: GAT model interpretation for AR prediction.

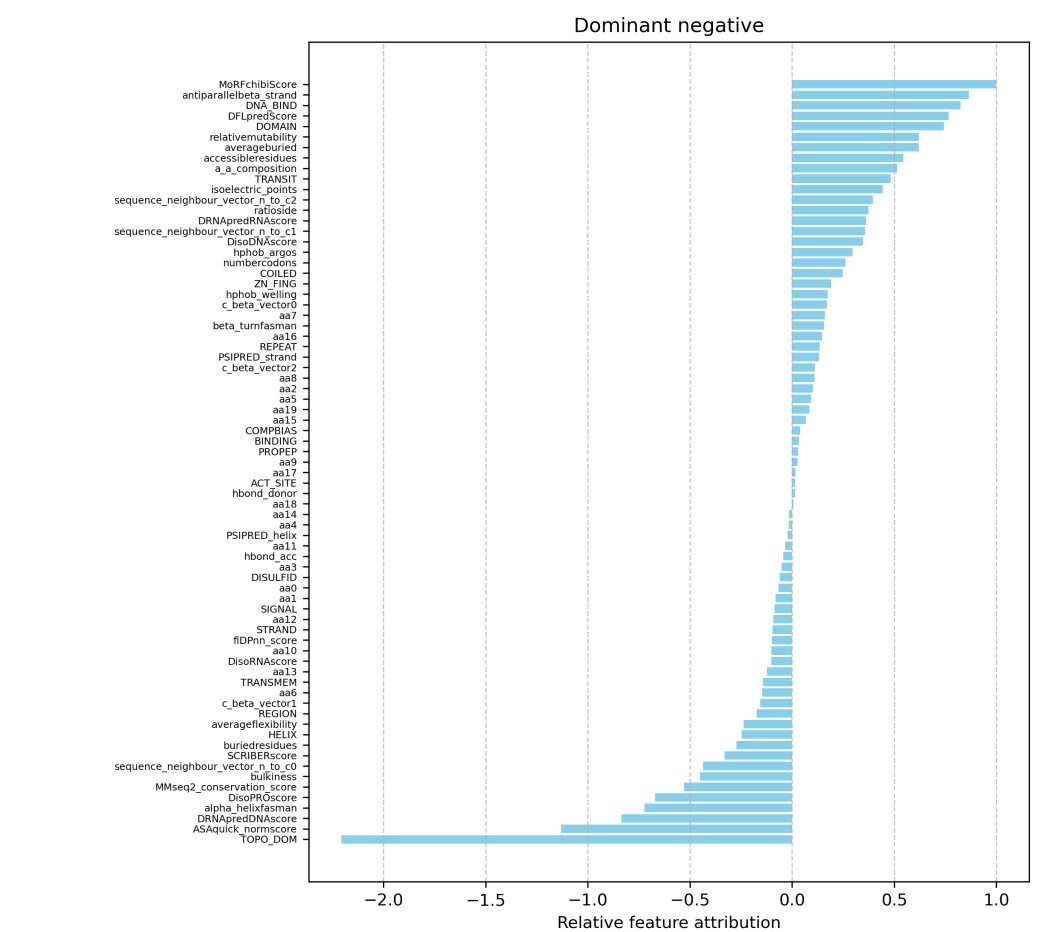

Supplementary Figure S3: GCN model interpretation for DN prediction.

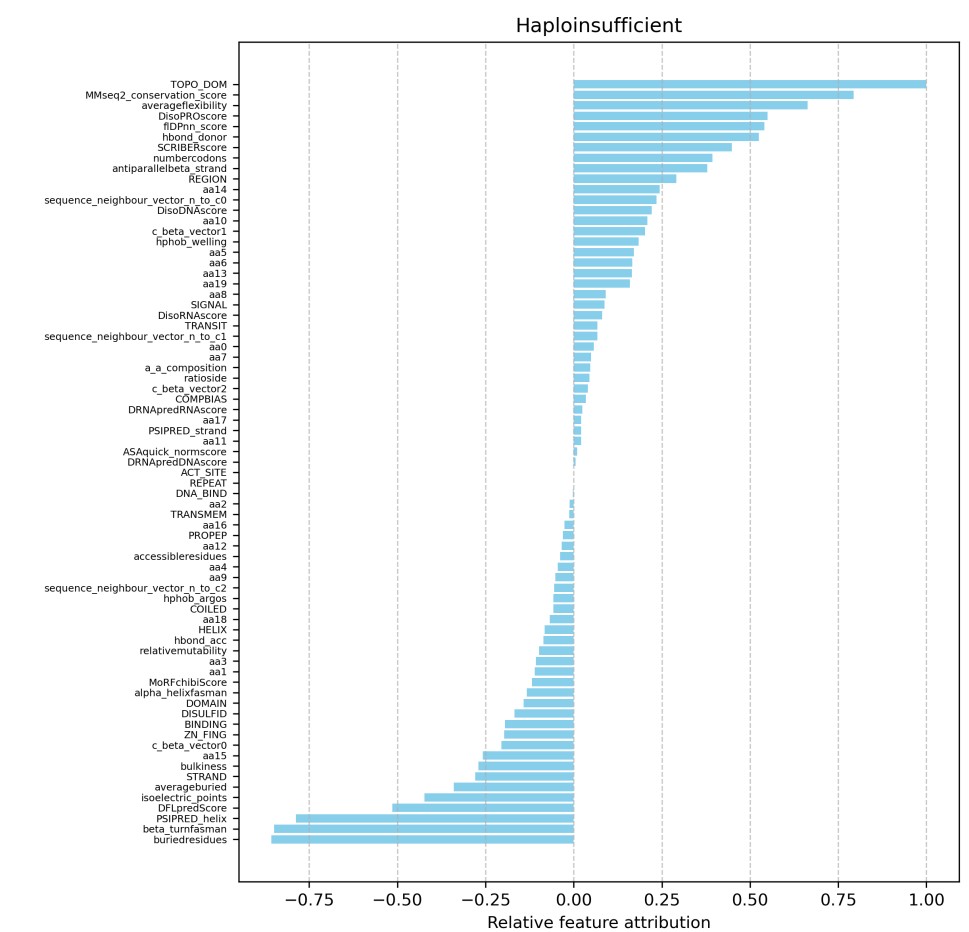

Supplementary Figure S4: GCN model interpretation for HI prediction.

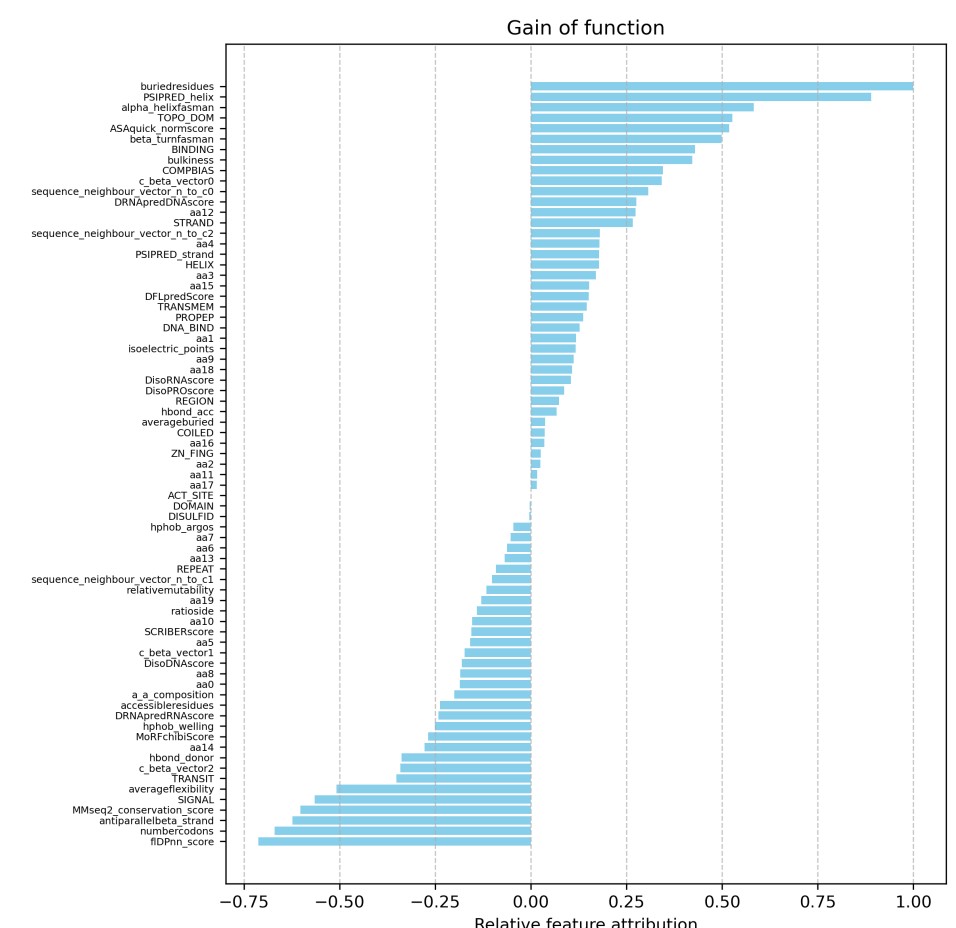

Supplementary Figure S5: GCN model interpretation for GOF prediction.

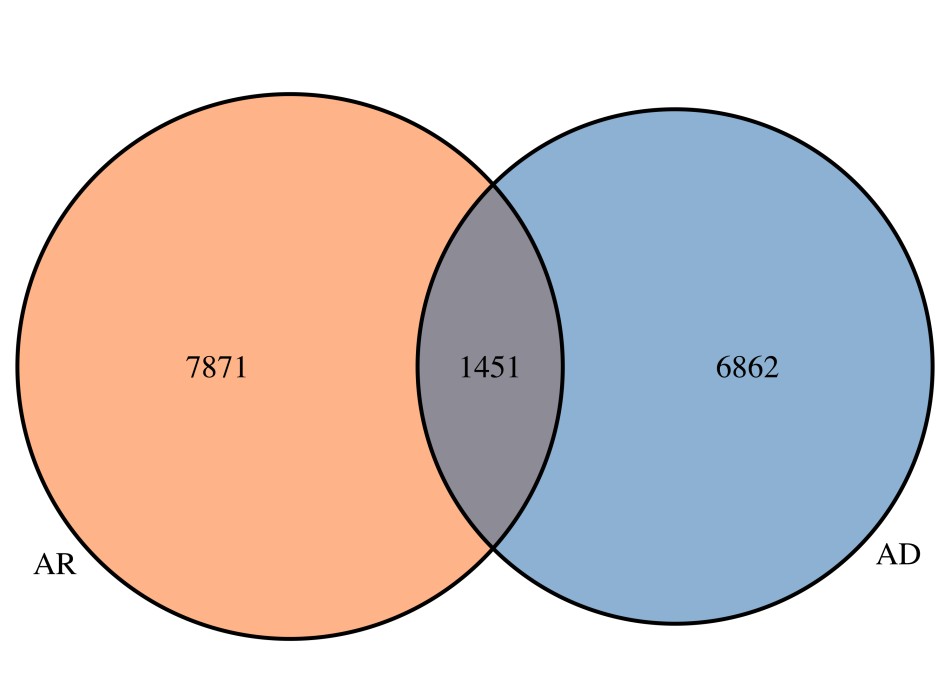

Supplementary Figure S6: Number of proteins with their MOI predictions.

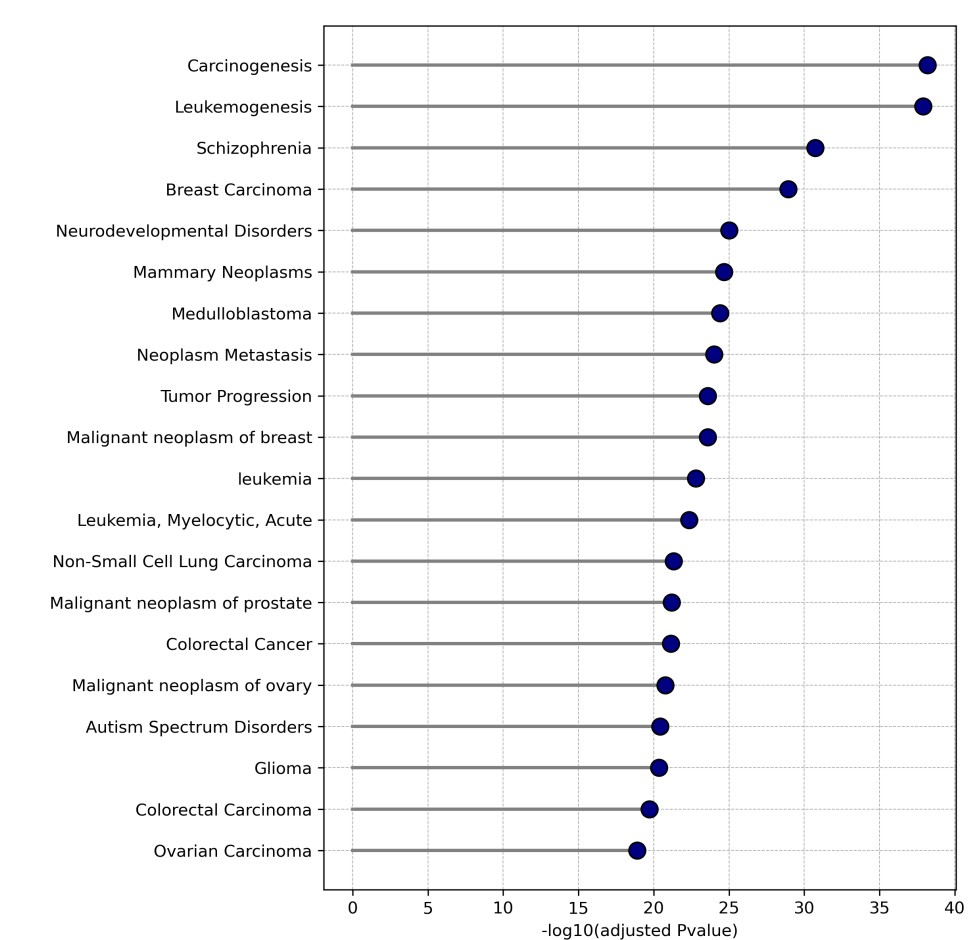

**Supplementary Figure S7:** Top 20 enriched diseases in AD proteins.

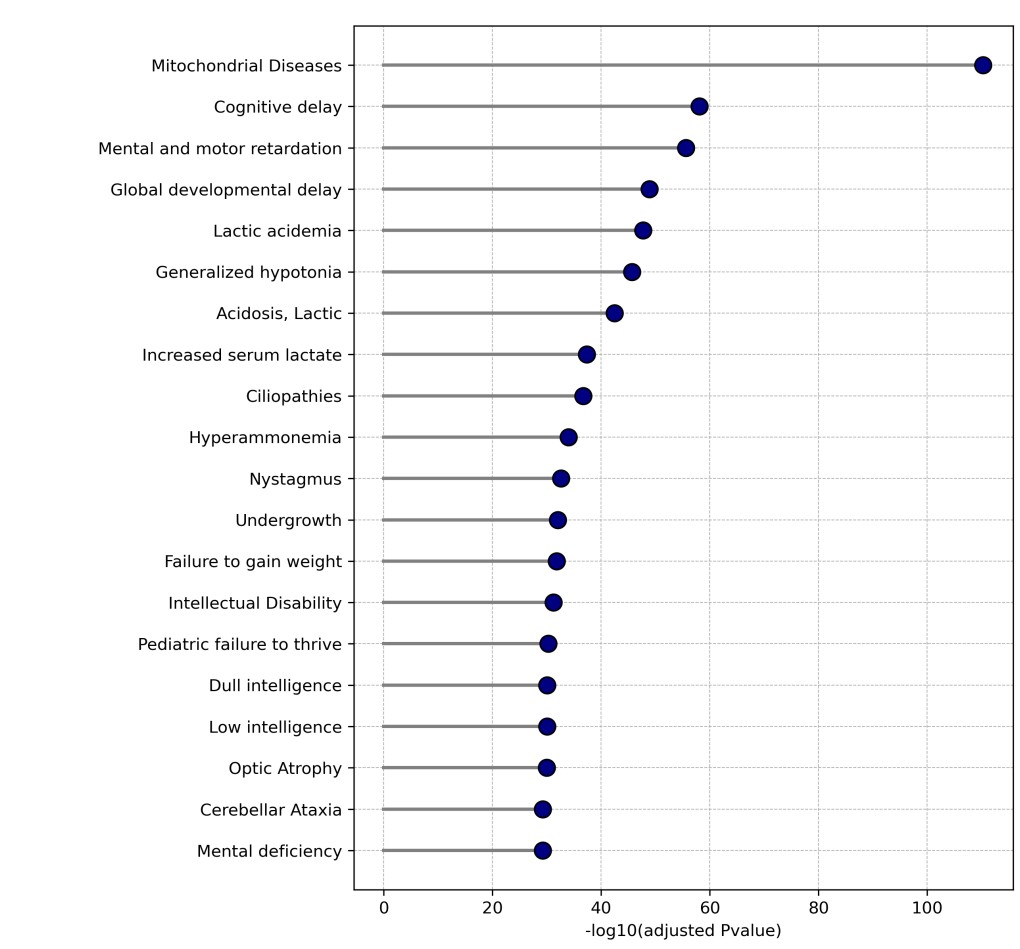

Supplementary Figure S8: Top 20 enriched diseases in AR proteins.

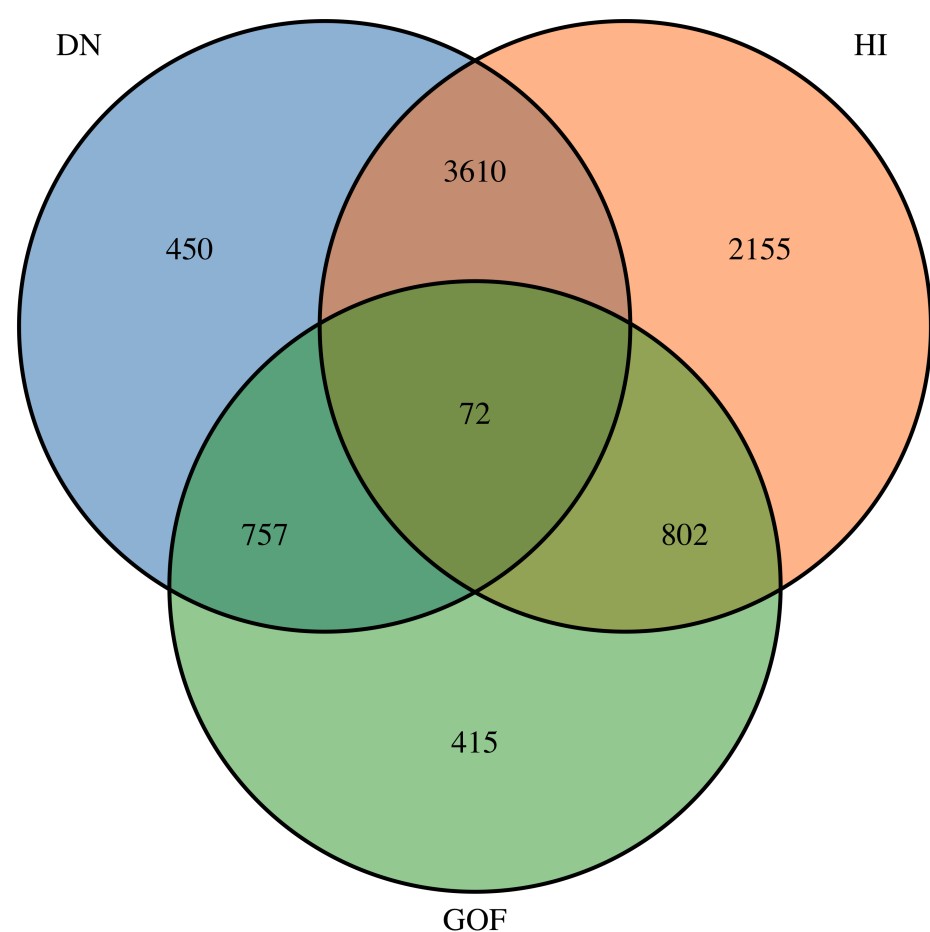

Supplementary Figure S9: Number of proteins with their molecular mechanism predictions.

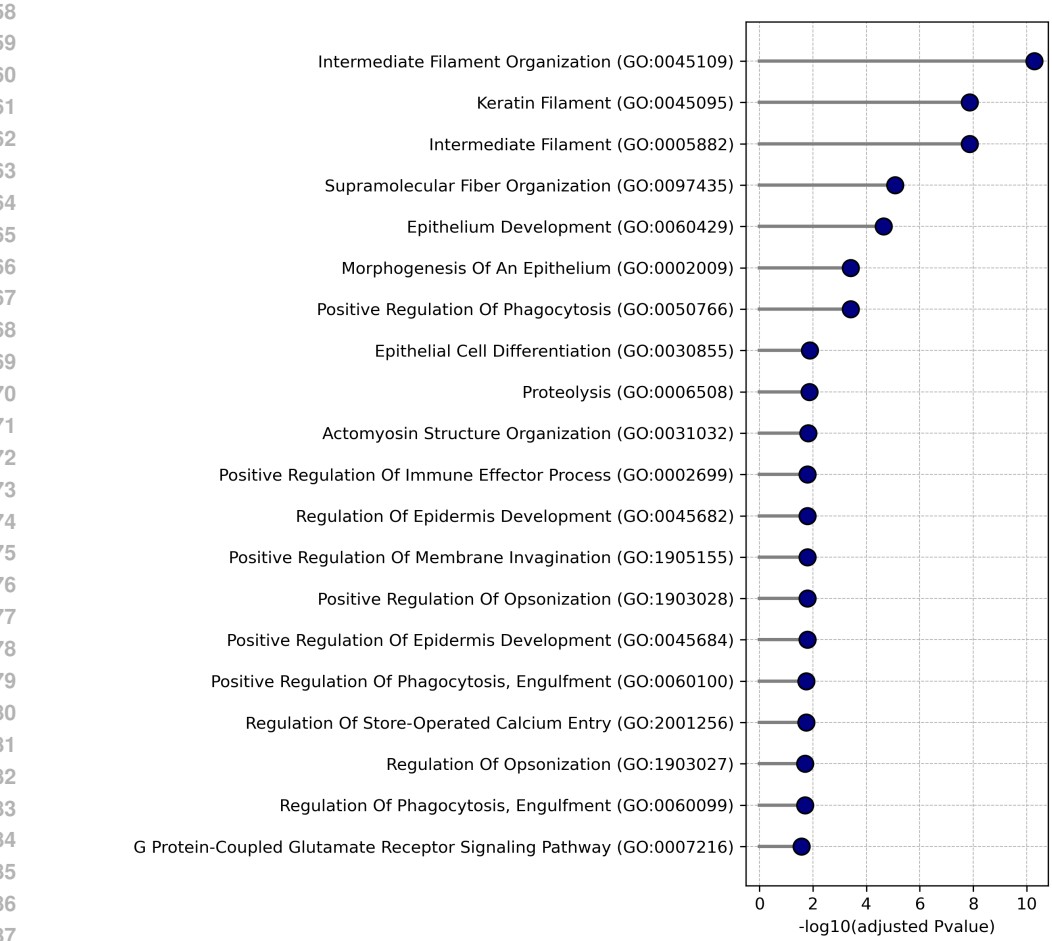

Supplementary Figure S10: Top 20 enriched pathways for DN proteins.

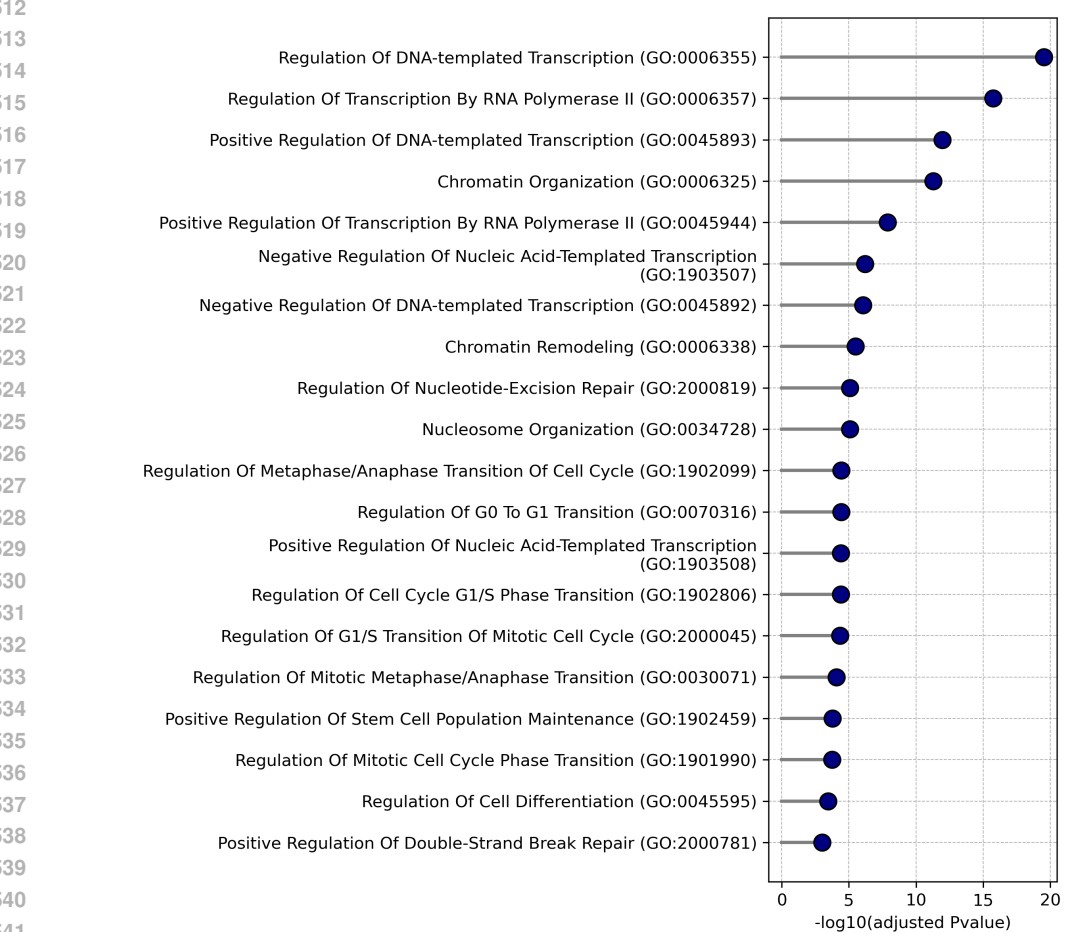

Supplementary Figure S11: Top 20 enriched pathways for HI proteins.

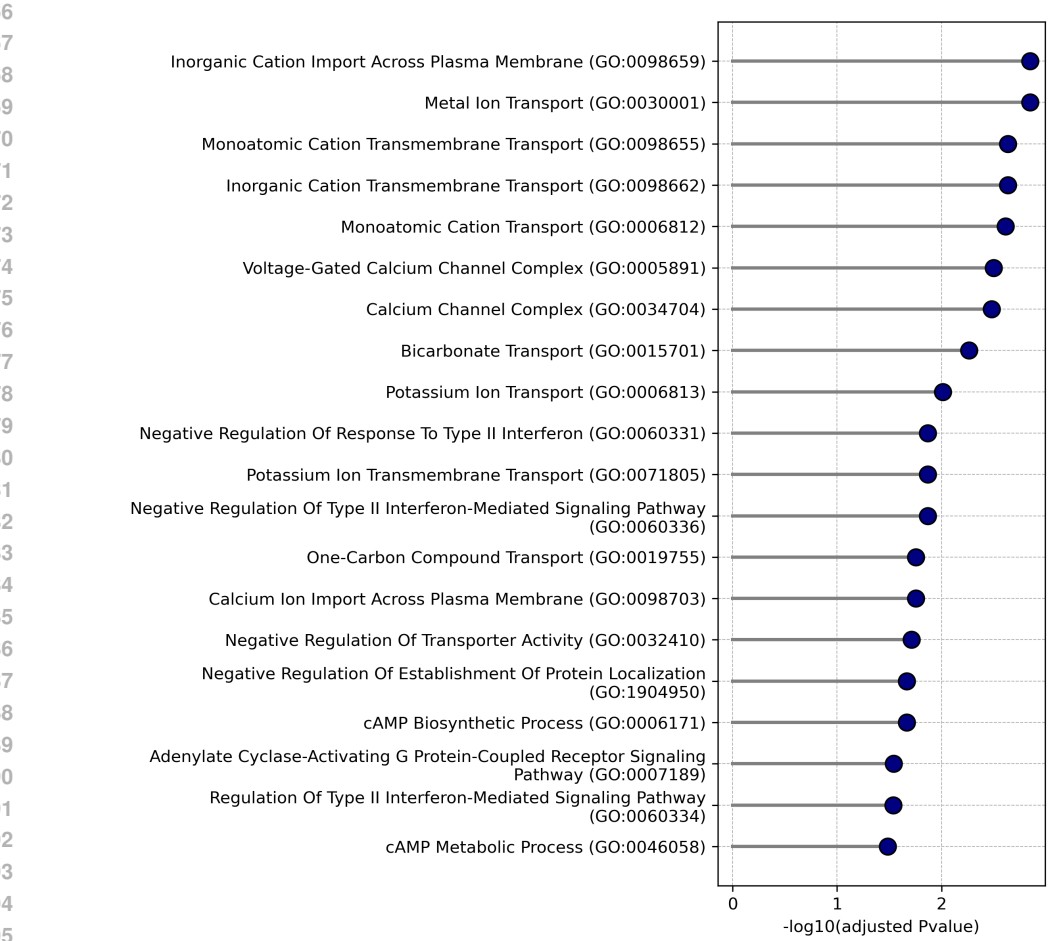

Supplementary Figure S12: Top 20 enriched pathways for GOF proteins.

