# OpenReview forum: "Proteome-wide prediction of mode of inheritance and molecular mechanism underlying genetic diseases using structural interactomics"
_ICLR.cc/2025/Conference — ICLR 2025 Conference Withdrawn Submission_

### Official Review · Reviewer_8LXJ · 2024-10-19

**Soundness:** 2
**Presentation:** 2
**Contribution:** 1
**Rating:** 3
**Confidence:** 3

**Summary:**

This paper gives a framework to predict the mode of inheritance of diseases and classify dominant-associated proteins based on their functional effect. The biggest highlight is its use of a graph-of-graphs idea to combine the protein-protein interaction networks and high-resolution protein structure.

**Strengths:**

The graph of graphs idea to predict the mode of inheritance of diseases is novel.

The methods are described in great detail and with persuasive experiments.

**Weaknesses:**

My biggest concern is that although this work seems to be relevant for predicting mode of inheritance and classifying functional effects, its contribution to deep learning models in the application domain of biology is insufficient. It is well known that GCN, GIN,GAT are three very classical GNN models.

And, the graph in graphs idea is also similar to the idea of the paper [1]. So, as far as ICLR is concerned, I think this may not be a notable paper for the community.

Also, I would suggest that the author modify the size of each figure to make the content and fonts in the figures look a little more harmonious.

[1] Gao Z, Jiang C, Zhang J, et al. Hierarchical graph learning for protein–protein interaction[J]. Nature Communications, 2023, 14(1): 1093.

**Questions:**

Please see the Weaknesses part. Thank you!

---

### Official Review · Reviewer_kSXk · 2024-11-02

**Soundness:** 2
**Presentation:** 1
**Contribution:** 1
**Rating:** 3
**Confidence:** 4

**Summary:**

The authors propose an approach to predict the likelihood for a protein to result in a disease if a mutation occurs on one of the inherited  copies using a graph neural networks method. They propose to use two scales to create a graph of graphs representation: at a protein level nodes are entire proteins and edges are interactions between proteins and at a residue level nodes are amino acids and edges are the type of bonds between these.

**Strengths:**

The integration of information at multiple scales is of interest.

**Weaknesses:**

The authors do not present a method capable to integrate information at various scales but rather work independently at each scale without exploiting any form of communication between scales.

**Questions:**

- the features for the nodes at each scale seem to be engineered and not learnable: is it true?
- couldn't one learn a graph encoding from the residue level and add it to the features at the protein interaction scale?
- all empirical results  are reported without a notion of dispersion; is it possible to repeat the experiments to get a measure of variance to understand the significance of the results?
- when comparing multiple approaches could you use a critical diagram of differences (e.g. https://scikit-posthocs.readthedocs.io/en/latest/generated/scikit_posthocs.critical_difference_diagram.html)
- Page 8 lines 417: why are the results notable? what would the enrichment analysis of a random set of protein yield instead? how about a non-random baseline, e.g. a nearest neighbour predictor.

---

### Official Review · Reviewer_F49n · 2024-11-02

**Soundness:** 2
**Presentation:** 3
**Contribution:** 2
**Rating:** 5
**Confidence:** 4

**Summary:**

The authors present a methodology able to detect both mode of inheritance (MOI) of proteins encoded by autosomal genes and the functional effects of gene variants. The strategy relies on established architectures like GCN, GAT, and GIN, using protein-protein interaction (PPI) data for MOI prediction (node classification) and protein structures obtained by AlphaFold for function prediction (graph classification). The author compared their method with two established strategies, one for MOI prediction (LDA) and one for functional effect prediction (SVM). The results reported by the authors show better metrics for their methodology. To inspect the biological validity of their results, the authors performed an enrichment analysis and determined the most influential features for the predictions via XAI.

**Strengths:**

The main strengths of the paper are the following:

1) The paper is well-written and easy to follow.

2) The problems addressed in the paper are relevant

3) It is extremely interesting to have a methodology able to address both MOI and functional effects prediction instead of needing to rely on two different strategies for the two tasks.

4) The bioinformatics-related work and processing is accurate.

5) The figures provided help convey the message of the authors more effectively.

**Weaknesses:**

The main weaknesses of the paper are the following:

1) The authors did not provide any code. This impinges on the reproducibility and further evaluation of their methods and results.

2) From the methodological point of view, there seems to be not much novelty. The authors use established architectures "out-of-the-box" to tackle the proposed tasks.

3) It seems that the authors did not perform any parameter tuning on their models. Additionally, no information on the hyperparameters used in the model is provided. The authors state they use dropout and weight decay, but no value for those hyperparameters is shown.

4) By reading the paragraph "Training and evaluation," it seems that the authors split the dataset into just two sets and not into training, validation, and test sets. They probably only used training and test sets if they did not perform hyperparameter tuning.

5) Regarding the explainability phase, IntegrateGradients was used to obtain global feature importance attributions by averaging the attributions of correctly predicted samples. I am not sure this is the correct approach to obtain global feature attributions. Leaving out the wrongly predicted samples from the averaging process may produce biased results. I suggest using global feature attribution methods instead. One example can be SAGE (Covert, Ian, Scott M. Lundberg, and Su-In Lee. "Understanding global feature contributions with additive importance measures." Advances in Neural Information Processing Systems 33 (2020): 17212-17223), among others.

6) Comparing against just one methodology per task (LDA and SVM) seems to me not enough to evaluate the performance of the strategy.

7) Given the tasks are multiclass classifications on unbalanced datasets, showing the results in terms of precision, recall, and F1 only without specifying the type of averaging strategy (micro, macro) used or without providing a confusion matrix conveys too little information to really understand the accuracy of the method (in particular class-wise).

8) The enrichment analysis reports only the enriched terms, but there are no links to the literature that confirm or better describe the association between the enriched terms and the proteins.

Overall, given the strong bioinformatics focus, I believe that after some revisions the paper can be accepted in a more specialized venue/journal, but given the limited methodological contribution and the flaws/imprecisions in model training and evaluation, I am afraid the work is not ready for publication in a high-impact machine-learning-focused conference at its current state.

**Questions:**

In order to improve the paper, the authors could perform hyperparameter tuning and give more information on the hyperparams used. They could compare against a higher number of methodologies and provide better ways to convey the results (confusion matrices may help). Morevoer, a literature search to verify the enriched terms could be performed.

---

### Note · Authors · 2024-11-14

**Comment:**

We would like to thank the reviewers for their feedbacks and comments.

**Withdrawal Confirmation:**

I have read and agree with the venue's withdrawal policy on behalf of myself and my co-authors.